# miR-328-3p targets TLR2 to ameliorate oxygen-glucose deprivation injury and neutrophil extracellular trap formation in HUVECs via inhibition of the NF-κB signaling pathway

**Mengting Yao**[1,2], **Chucun Fang**[1,2], **Zilong Wang**[1,2], **Tianting Guo**[3], **Dongwen Wu**[1,2], **Jiacheng Ma**[1,2], **Jian Wu**[1,2], **Jianwen Mo**[2]*

**1** Gannan Medical University, Ganzhou, Jiangxi, China, **2** Department of Orthopedic Surgery, The First Affiliated Hospital of Gannan Medical University, Ganzhou, Jiangxi, China, **3** Department of Orthopedics, Guangdong Provincial People's Hospital Ganzhou Hospital, Ganzhou Municipal Hospital, Ganzhou, Jiangxi, China

* mjw1997@126.com

**Data Availability Statement:** All relevant data are within the manuscript and its Supporting information files.

## Abstract

### Background

Endothelial cell injury is one of the important pathogenic mechanisms in thrombotic diseases, and also neutrophils are involved. MicroRNAs (miRNAs) have been demonstrated to act as essential players in endothelial cell injury, but the potential molecular processes are unknown. In this study, we used cellular tests to ascertain the protective effect of miR-328-3p on human umbilical vein endothelial cells (HUVECs) treated with oxygen-glucose deprivation (OGD).

### Methods

In our study, an OGD-induced HUVECs model was established, and we constructed lentiviral vectors to establish stable HUVECs cell lines. miR-328-3p and Toll-like receptor 2 (TLR2) interacted, as demonstrated by the dual luciferase reporter assay. We used the CCK8, LDH release, and EdU assays to evaluate the proliferative capacity of each group of cells. To investigate the expression of TLR2, p-P65 NF-κB, P65 NF-κB, NLRP3, IL-1β, and IL-18, we employed Western blot and ELISA. Following OGD, each group's cell supernatants were gathered and co-cultured with neutrophils. An immunofluorescence assay and Transwell assay have been performed to determine whether miR-328-3p/TLR2 interferes with neutrophil migration and neutrophil extracellular traps (NETs) formation.

### Results

In OGD-treated HUVECs, the expression of miR-328-3p is downregulated. miR-328-3p directly targets TLR2, inhibits the NF-κB signaling pathway, and reverses the proliferative

**Funding:** his work was supported by the National Natural Science Foundation of China (No. 82160375); Natural Science Foundation of Jiangxi Province (No. 20202BABL206035); Science and Technology Planning Project of Jiangxi Provincial Health Commission in 2023 (NO. 202312146); Science and Technology Program of Jiangxi Provincial Administration of Traditional Chinese Medicine (Project No. 2021A374); and Ganzhou City Science and Technology Plan Project (NO. 2023LNS26841). The funders had no role in study design, data collection and analysis, decision to publish, or preparation of the manuscript.

**Competing interests:** The authors have declared that no competing interests exist.

capacity of OGD-treated HUVECs, while inhibiting neutrophil migration and neutrophil extracellular trap formation.

## Conclusions

miR-328-3p inhibits the NF-κB signaling pathway in OGD-treated HUVECs while inhibiting neutrophil migration and NETs formation, and ameliorating endothelial cell injury, which provides new ideas for the pathogenesis of thrombotic diseases.

## 1. Introduction

It is generally recognized that Virchow's triad is the main factor contributing to thrombosis [1], in which vascular endothelial cell injury occupies an important position [2]. Many studies have shown that ischemia and hypoxia can trigger an inflammatory response, which is an essential pathological cause of endothelial cell injury. Ischemia and hypoxia activate multiple pathways that damage endothelial cells, while neutrophils accumulate at the site of injury [3], exacerbating the inflammatory response. Despite considerable progress in understanding the mechanisms of endothelial cell injury in thrombotic diseases, effective biomarkers to reverse ischemia-hypoxia-induced cell injury are still lacking. Therefore, it is crucial to focus on the inflammatory response-related mechanisms behind endothelial cell injury, which may facilitate the development of novel drugs.

Neutrophils are derived from hematopoietic stem cells of the bone marrow, which are chemotactic, phagocytic, and bactericidal and are important in nonspecific cellular immunity. Large extracellular reticular structures known as neutrophil extracellular traps (NETs) are released by neutrophils, in response to inflammatory stimuli or hypoxia, etc [4]. Dominated by chromatin DNA, they contain a variety of enzymes and proteins, especially Myeloperoxidase (MPO), Neutrophil Elastase (NE), Histone G, Histone H3 Citrulline (H3Cit), and neutrophil antimicrobial proteins [5]. Upon stimulation, neutrophils become active and have the ability to damage endothelium cells. In addition, NETs can be activated by activated endothelial cells, and NETs produce endothelial cytotoxicity. Neutrophil and endothelial cell activation form a vicious circle. Thus, neutrophil migration and the NETs formation are closely associated with the mechanism of endothelial cell injury.

MicroRNAs (miRNAs), which influence post-transcriptional gene regulation, consist of 18–25 nucleotides and are small, highly conserved non-coding RNAs. MiRNAs can bind to the 3'-UTR of target genes to inhibit the protein levels of target genes thereby regulating cell biological functions such as cell proliferation, differentiation, apoptosis, and disease progression [6]. For example, it was discovered that overexpressing miR-126 inhibited oxidative stress and inflammation in HUVECs, a process that primarily involves activation of the SIRT1/Nrf2 signaling pathway [7]. According to Zhang et al., when H9c2 cells are hypoxic, interfering with microRNA-495 prevents apoptosis via targeting NFIB [8]. It has recently been found that miR-328-3p affects the beginning and progression of diseases, particularly in a range of malignancies [9–11]. Nonetheless, it is still unclear how miR-328-3p prevents vascular endothelial cell damage in ischemia-hypoxic environments.

Toll-like receptors (TLRs) are single transmembrane non-catalytic proteins that have been implicated in a range of inflammations and diseases, primarily recognizing pathogens and participating in damage-associated molecular patterns of innate immunity [12]. It has been found that TLR2 is abundant in endothelial cells in a variety of tissues and is required for pro-

inflammatory endothelial cell function. TLR2 is closely related to the occurrence of endothelial cell inflammation and injury [13, 14]. TLR2 dimerization can trigger the nuclear factor-kappa B (NF-κB) pathway, which specifically binds DNA and regulates protein transcription [15], which triggers the transcription of various target genes and controls and participates in cell cycle progression, inflammatory responses, and inhibition of cell adhesion and apoptosis [16]. More research is necessary to fully understand how miR-328-3p regulates vascular endothelial cell damage by influencing the NF-κB pathway via TLR2.

Therefore, to simulate the ischemic-hypoxic environment in vitro, the oxygen-glucose deprivation approach was employed in this study. We explored the protective role of miR-328-3p against inflammatory injury in HUVECs through the TLR2 /NF-κB signaling pathway, as well as the role of neutrophils in this process. The significance of miR-328-3p in vascular endothelial cells and novel insights into the cellular molecular mechanisms of thrombotic disorders are both supported by our findings.

## 2. Materials and methods

### 2.1 Cell culture

The medium used for cell culture was Dulbecco's Modified Eagle Medium (DMEM, C11995500BT, Gibco, China), which was enhanced with 1% penicillin-streptomycin (15140–122, Gibco, USA) and 5% fetal bovine serum (FBS, A3160801, Gibco, UK). HUVECs (FH1122, STR 20170721–05, Fuheng Biology, Shanghai, China) were cultivated in an incubator with 5% $CO_2$ and a temperature of 37˚C. When the cells were fused to 80%, we digested and passaged them with 0.25% Trypsin-EDTA (25200–056, Gibco, Canada). Tests were performed after the cell traits were stabilized by 3 consecutive passages.

### 2.2 Cell transfection

Introduction of lentivirus (miR-328-3p mimic, miR-328-3p inhibitor, negative control mimic, and negative control inhibitor) (GenePharma, China) into cells and screening by Puromycin Solution (BL528A, Biosharp, China) to obtain stably expressing cell lines. The vector we used were all LV3 (H1/GFP&Puro) and the sequences are shown in Table 1.

### 2.3 Oxygen-glucose deprivation (OGD) model

Glucose-free DMEM (D6540, Solarbio, China) was added after washing the cells once with $1 \times$ phosphate buffered solution (PBS). To reach anaerobic conditions, groups of cells were placed in an anaerobic incubator for 30 min to reduce the oxygen concentration to 1%. We maintained conditions of 94% N2, 5% CO2, and 1% O2, and after 6h of oxygen-glucose deprivation, the basal medium was added in place of Glucose-free DMEM. For a full day, the cells were grown in standard conditions. A total of 6 groups were obtained: Normal, OGD-Normal, OGD-NC mimic, OGD-mimic, OGD-NC inhibitor, and OGD-inhibitor.

**Table 1. Sequences for lentivirus transfection.**

| Name | Sequence (5'-3') |
| --- | --- |
| miR-328-3p mimic | CTGGCCCTCTCTGCCCTTCCGT |
| negative control mimic | TTCTCCGAACGTGTCACGT |
| miR -328-3p inhibitor | ACGGAAGGGCAGAGAGAGGGCCAG |
| negative control inhibitor | TTCTCCGAACGTGTCACGT |

## 2.4 Cell Counting Kit-8 (CCK-8) assay

$2\times10^3$ cells were inoculated in 96-well plates. After OGD, we used the cell Counting kit -8 kit (C0038, Beyotime Biotechnology, Shanghai, China) to detect cell proliferation. After adding reagents, place in an incubator at 37˚C for 1.5 hours.

## 2.5 Lactate dehydrogenase (LDH) releases assay

Membrane integrity and cell viability can be determined by LDH release assays, and we use the LDH assay kit (A020-2-2, Nanjing Jiancheng Bioengineering Institute, China) to detect cellular damage. After OGD, we collected the cell supernatants, operated according to the kit instructions, and measured the absorbance value at 450nm using a microplate reader.

## 2.6 5-Ethynyl-2´-deoxyuridine (EdU) assay

We inoculated each group of cells at $2\times10^3$ per well in a 96-well plate. After OGD, we added the EdU working solution (G1603, Servicebio, China) and then used 4% paraformaldehyde (BL539A, Biosharp, China) and 0.5% Triton (P0096, Beyotime Biotechnology, Shanghai, China) according to the instructions. We treated the cells with Apollo reaction solution for 30 min under light-avoidance conditions. After staining with Hoechst33342, 3 random fields were taken under a fluorescence microscope. The ratio of red fluorescence (EdU-stained cells) to blue fluorescence (Hoechst 33,342-stained cells) in the obtained image indicates the percentage of EdU-positive cells.

## 2.7 Dual-luciferase reporter assay

The Dual-luciferase reporter assay was employed to identify possible binding between miR-328-3p and the 3'-UTR of TLR2. Wild-type (WT) or mutant (MUT) TLR2 3'-UTR sequences were inserted into the vectors, and after transfection for a specific time, we measured luciferase activity and normalized it (Promega, USA). The difference between the WT and MUT 3'-UTR sequences is that the WT is base complementary paired with the miRNA, the MUT is not. The vector we used were all psiCHECK™-2-Vector (GenePharma, China) and the sequences are shown in Table 2.

## 2.8 Quantitative real-time PCR (qRT-PCR)

TRIzol (R0016, Beyotime Biotechnology, Shanghai, China) was used to extract total RNA from six groups of cells, and RNA concentration and purity were measured by UV spectrophotometer. To analyze miR-328-3p expression, we synthesized cDNA using the stem-loop method (B532453, Sangon Biotech, China). The real-time RT-PCR was performed using the micro-RNAs qPCR Kit (SYBR Green Method) (B639271, Sangon Biotech, China) according to the instructions, using U6 as Endogenous Reference Genes Primers. For the analysis of TLR2, TLR3, TLR4, TLR9, NF-κB, NLRP3, IL-1β, and IL-18, a PrimeScript™ RT reagent Kit with gDNA Eraser (Perfect Real Time) (Cat# RR047A, Takara, China) was used to synthesize cDNA. Using β-actin as the Endogenous Reference Genes Primers, cDNA was evaluated using TB Green® Premix Ex Taq™ II (Tli RNaseH Plus) (Cat# RR820A, Takara, China) for real-time

**Table 2. Sequences for dual-luciferase reporter assay.**

| Name | Sequence (5'-3') |
|---|---|
| hsa-TLR2-miR-328-3p-WT | ggttgacttcatggatgcagaacccatggatatagAGGGCCAactgtaatctgtagcaactggcttagt |
| hsa-TLR2-miR-328-3p-MUT | ggttgacttcatggatgcagaacccatggatatagTCCCGGTactgtaatctgtagcaactggcttagt |

**Table 3. Primer sequences for qRT-qPCR.**

| Gene | Forward 5'-3' | Reverse 5'-3' |
|---|---|---|
| miR-328-3p | AACAATCTGGCCCTCTCTGC | CAGTGCAGGGTCCGAGGT |
| U6 | CTCGCTTCGGCAGCACA | AACGCTTCACGAATTTGCGT |
| TLR2 | AAGCAGCATATTTTACTGCTGG | CCTGAAACAAACTTTCATCGGT |
| TLR3 | CCTGGTTTGTTAATTGGATTAACGA | TGAGGTGGAGTGTTGCAAAGG |
| TLR4 | TTTGGACAGTTTCCCACATTGA | AAGCATTCCCACCTTTGTTGG |
| TLR9 | TGGTGTTGAAGGACAGTTCTCTC | CACTCGGAGGTTTCCCAGC |
| P65 NF-κB | AACAGAGAGGATTTCGTTTCCG | TTTGACCTGAGGGTAAGACTTCT |
| NLRP3 | CGTGAGTCCCATTAAGATGGAGT | CCCGACAGTGGATATAGAACAGA |
| IL-1β | ATGATGGCTTATTACAGTGGCAA | GTCGGAGATTCGTAGCTGGA |
| IL-18 | GCTGAAGATGATGAAAACCTGG | CAAATAGAGGCCGATTTCCTTG |
| β-actin | CCTGGCACCCAGCACAAT | GGGCCGGACTCGTCATAC |

RT-PCR. Each sample was replicated three times. We used the $2^{-\Delta\Delta Ct}$ method to calculate qRT-PCR Ct values. All primers (Table 3) were ordered from Sangon Biotech.

## 2.9 Protein extraction and Western blot analysis

We collected six sets of cell samples and lysed proteins with RIPA buffer (R0010, Solarbio, China) that contained Aprotinin and PMSF. Protein concentrations were determined by the BCA method, SDS-PAGE electrophoresis was performed using 7.5% and 12.5% separation gels (PG111, PG113, Epizyme Biotech, China), respectively, and the samples were transferred to PVDF membranes. Following blocking, we incubated the membranes with primary antibodies against TLR2 (Catalog No. 66645-1-Ig, Proteintech, China), P65 NF-κB (Catalog No. 10745-1-AP, Proteintech, China), p-P65 NF-κB (Cat.#: AF2006, Affinity Biosciences, China), NLRP3 (#M035175, abmart, China), IL-1β (Catalog No. 16806-1-AP, Proteintech, China), IL-18 (Catalog No. 60070-1-Ig, Proteintech, China), and β-actin (Catalog No. 20536-1-AP, Proteintech, China) at 4°C overnight. After washing in TBST, we incubated for 1 h using secondary HRP antibodies (Catalog No. SA00001-1, Catalog No. SA00001-2, Proteintech, China). The samples were exposed to a developer using ECL Plus reagent (P0018S, Beyotime Biotechnology, Shanghai, China). Grey-scale values were analyzed using Images-J to quantify protein expression levels based on band density.

## 2.10 Extraction of neutrophils

Peripheral whole blood was taken from healthy adults and anticoagulated with EDTA. Neutrophils were isolated using Polymorphprep (AS1114683, Oslo, Norway) and centrifuged at 500g for 30 minutes, following the isolation procedure as instructed. Carefully extract neutrophils, wash twice with PBS resuspension, and centrifuge at 400g for 10 minutes. Add erythrocyte lysis solution (R1010, Solarbio, China) and lyse at 4°C for 15 minutes, centrifuge and discard supernatant until no red color is present in the precipitate. We use a cell counter to count the number of cells. Healthy adults were recruited from March 1 to April 1, 2023, and all received written informed consent. The first affiliated Hospital of Gannan Medical University's Institutional Review Board and Ethics Committee approved the study (Ganzhou China, ethical approval No. LLSC-2023022801).

## 2.11 Transwell migration assay

We inoculated each group of cells in 24-well plates and collected cell supernatants after OGD. After centrifugation, we added the supernatant into a new 24-well plate and then placed trans-well chambers with a membrane pore size of 8 μm. Neutrophils were diluted to $10^6$ cells per ml and 200 μl was aspirated and added to the upper chamber. After 12 hours, the cytosol of the lower chamber was collected and counted using a cell counter.

## 2.12 Cell immunofluorescence

The number of neutrophils was set at $4 \times 10^6$ cells per ml, and 100 μl of each well was inoculated into 24-well plates with Polylysine slides. We collected cell cultures from each group after OGD. After centrifugation, we collected 400 μl of supernatants and added them to 24-well plates that were seeded with neutrophils and incubated in a 5% CO2, 37°C incubator. After overnight incubation, 4% paraformaldehyde was added slowly along the plate wall in each well. After permeabilizing by 0.5% Triton at room temperature, we blocked the non-specific antigen with the Goat Blocking Buffer. We mixed equal amounts of H3Cit (ab5013, Abcam) and MPO (Catalog No. 66177-1-Ig, Proteintech, China) primary antibodies, and added the primary antibody mixture to each well. After incubating at 4°C overnight, we added the secondary antibody (Catalog No. SA00013-1, Catalog No. SA00013-4, Proteintech, China) and incubated it for 1 hour in a wet box protected from light. Finally, stained with DAPI solution and observed by fluorescence microscope.

## 2.13 Enzyme-linked immunosorbent assay (ELISA)

We collected the supernatant after the different treatments. The instructions for the ELISA assay were followed in order to measure the levels of IL-1β (KE00021, Proteintech, China), IL-18 (KE00193, Proteintech, China), MPO (KE00171. Proteintech, China), and NE (JL12352, Jonln, China).

## 2.14 Statistical analysis

We use means ± standard deviation (SD) to present all data. To compare data between the two independent groups, we employ Student's t-test. SPSS 21.0 statistical software (IBM, USA) and GraphPad Prism 8.0 (La Jolla, USA) were used to analyze the experimental data and picture processing. A statistically significant difference was defined as $P < 0.05$.

# 3. Result

## 3.1 In HUVECs treated with OGD, overexpression of miR-328-3p reverses damage

According to the qRT-PCR results, OGD-treated HUVECs expressed less miR-328-3p than normal culture did, indicating that OGD treatment significantly reduced the expression of miR-328-3p. Following transfection, the qRT-PCR findings demonstrated a substantial change in the expression of miR-328-3p between the OGD-mimic and OGD-inhibitor groups and the OGD-NC mimic and OGD-NC inhibitor groups (Fig 1A). CCK8, LDH release, and EdU assays were used to assess the cell proliferative capacity in each group. The results of CCK8 and EdU assay showed that OGD treatment was able to reduce cell viability compared to the Normal group. The OGD-mimic group's ability for cell proliferation was higher than that of the OGD-NC mimic group; in contrast, the OGD-inhibitor group displayed the opposite trend (Fig 1B, 1C and 1E). The LDH release assay showed that OGD treatment could promote

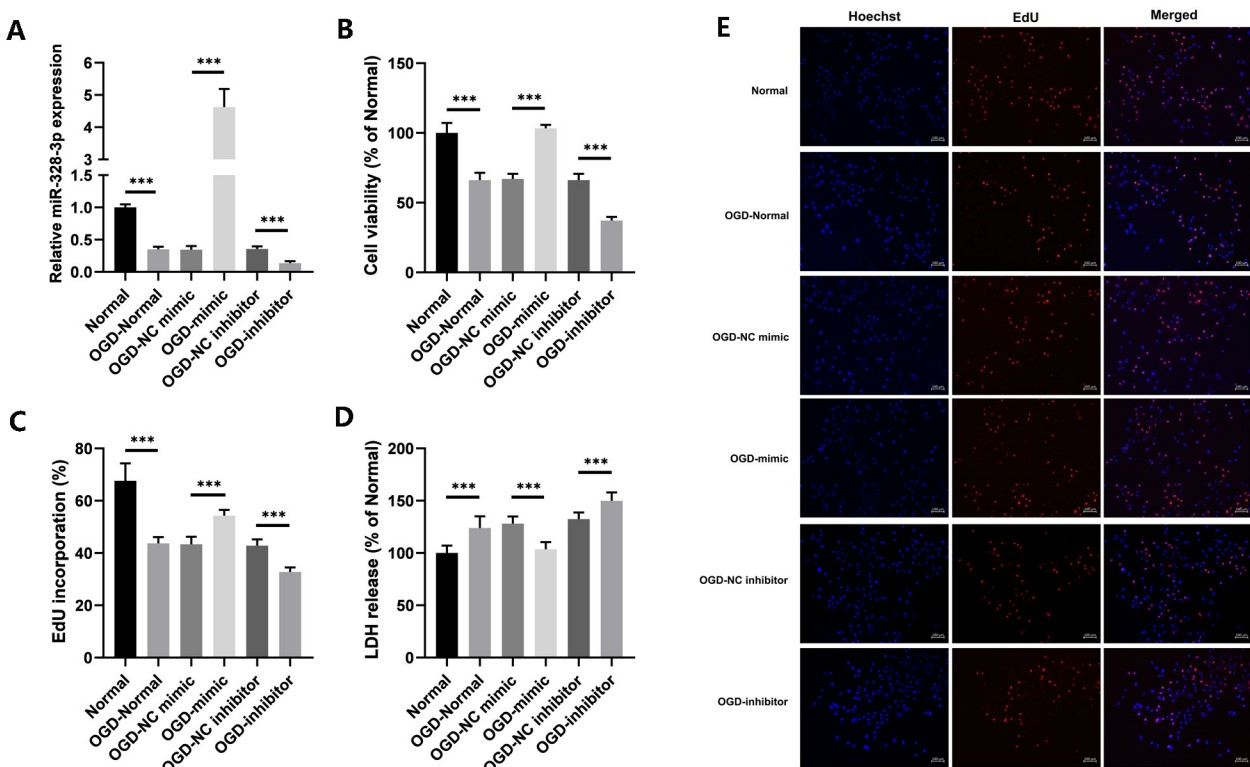

**Fig 1. miR-328-3p expression is down-regulated and promotes cell proliferation in OGD-treated HUVECs.** A. qRT-qPCR was used to detect the expression of miR-328-3p in OGD-treated cells in each group. B. Cell viability of each group was detected by CCK8 assay. C and E. EdU assays were performed to measure the effect of miR-328-3p on the proliferation of OGD-treated HUVECs. D. Detection of LDH release from OGD-treated cells in each group. Data are presented as the mean ± SD (n = 3). ***P < 0.001.

cellular LDH release. The percentage of LDH release in the OGD-mimic or OGD-inhibitor groups was significantly reduced or increased compared to the OGD-NC mimic or OGD-NC inhibitor groups (Fig 1D). In conclusion, following OGD treatment, miR-328-3p expression was decreased in HUVECs. In the meantime, HUVECs damage was reversed and cell viability was increased by miR-328-3p overexpression.

### 3.2 The miR-328-3p targets TLR2 in HUVECs treated with OGD and affects NF-κB signaling pathway expression as well as inflammation-related molecules

TargetScan was used to determine whether TLR2 is a potential target of miR-328-3p (Fig 2A). The Dual-luciferase reporter assay can detect whether miRNA and target protein are potentially bound. We discovered that miR-328-3p binds to the 3'-UTR of TLR2, and the 3'-UTR of TLR2 is a potential binding site for miR-328-3p (Fig 2B). OGD-treated HUVECs also showed dramatically elevated the protein and mRNA levels of TLR2, as demonstrated by Western blotting and qRT-qPCR. The TLR2 expression was markedly reduced or enhanced in the OGD-mimic group or OGD-inhibitor group compared with the OGD-NC mimic group or OGD-NC inhibitor group (Fig 2C and 2D). In addition, under the influence of OGD and miR-328-3p, we examined the expression of TLR3, TLR4, and TLR9 by qRT-PCR experiments. It was found that OGD treatment could affect the expression of TLR3, TLR4, and TLR9

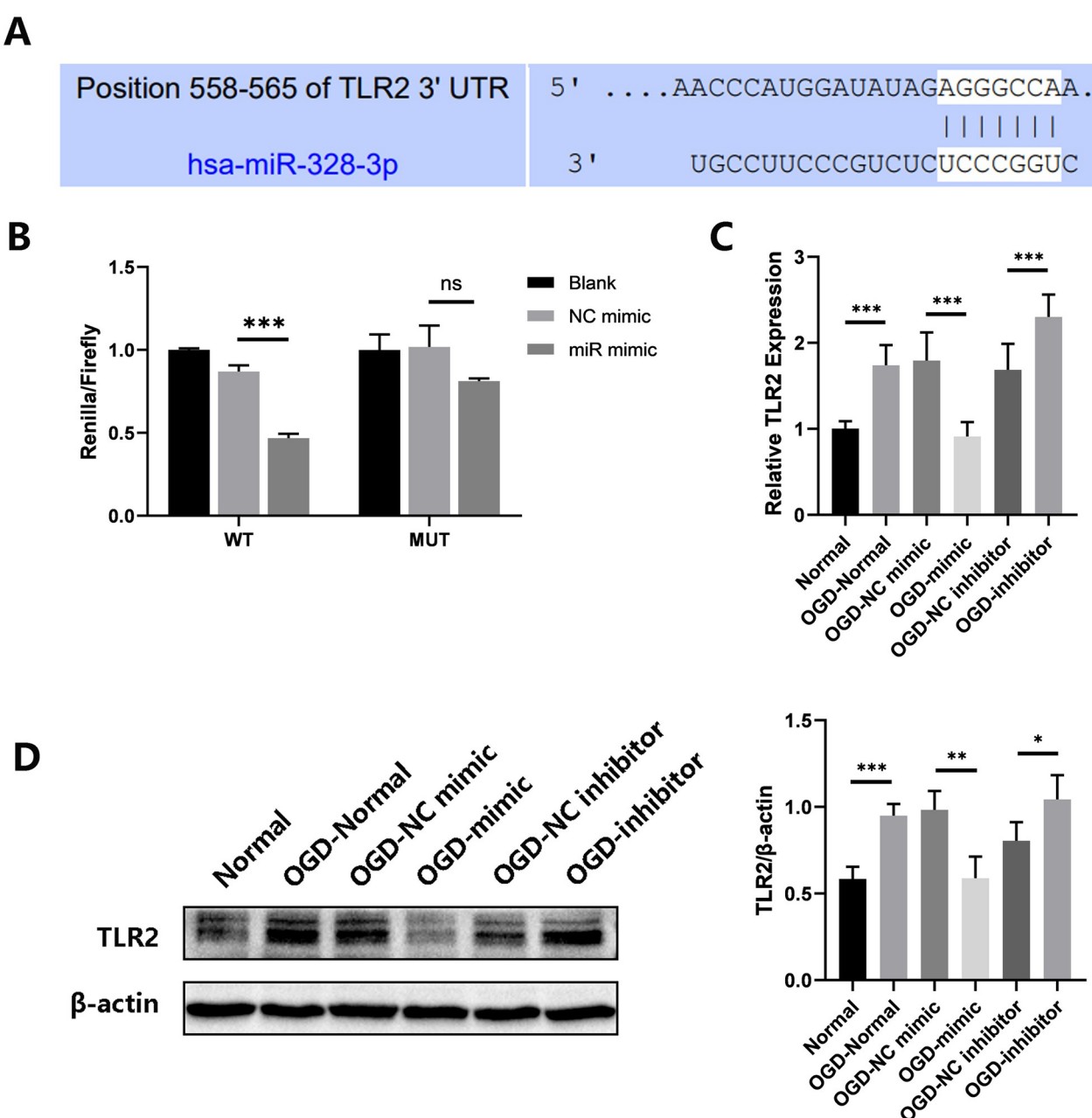

**Fig 2. miR-328-3p targets TLR2 and down-regulates TLR2 expression after OGD treatment.** A. 3'-UTR of TLR2 mRNA was predicted to be targeted by miR-328-3p. B. Dual-luciferase reporter assay detects the binding of TLR2 and miR-328-3p (n = 3). C. qRT-qPCR analyzed the mRNA expression level of TLR2 in each group (n = 3). D. Western blotting analyzed the protein expression level of TLR2 in each group (n = 4). Data are presented as the mean ± SD. **P < 0.01; ***P < 0.001; ns, no significance.

in HUVECs. However, there was no correlation trend of TLR3, TLR4, and TLR9 expression in HUVECs overexpressing and interfering with miR-328-3p (Fig 3). Based on the above experiments, we detected the protein expression of NF-κB and inflammation-related molecules. The OGD-Normal group had higher levels of p-P65 NF-κB, NLRP3, IL-1β, and IL-18 expression than the Normal group, according to the results of Western blotting. However, protein expression was reduced or enhanced in the OGD-mimic group or OGD-inhibitor group compared

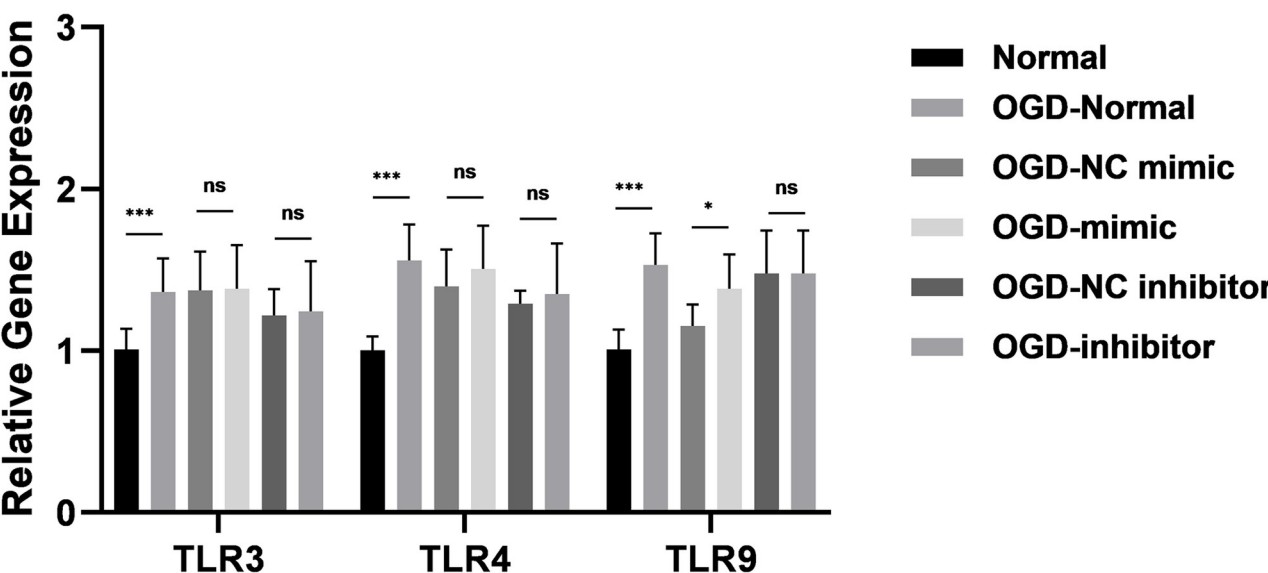

**Fig 3. OGD treatment up-regulates the expression of TLR3, TLR4, and TLR9, but is not associated with miR-328-3p.** qRT-qPCR analyzed the mRNA expression level of TLR3, TLR4, and TLR9 in each group (n = 3). Data are presented as the mean ± SD. **P < 0.01; ***P < 0.001; ns, no significance.

to the OGD-NC mimic group or OGD-NC inhibitor group (Fig 4A). The qRT-qPCR results showed the same trend (Fig 4B). According to the results above, miR-328-3p binds TLR2 and suppresses TLR2 expression in OGD-treated HUVECs while influencing the expression of NF-κB signaling pathway and inflammation-related molecules.

### 3.3 Overexpression of miR-328-3p attenuates neutrophil migration after OGD treatment

It is established that IL-1β and IL-18 mediate neutrophil recruitment [17] and amplify inflammatory responses. In order to investigate the impact of miR-328-3p overexpression on neutrophils under OGD induction, we used ELISA to identify IL-1β and IL-18 secretion in the cell supernatants after OGD treatment. The OGD-Normal group released more IL-1β and IL-18 than the Normal group, and the results are identical to the Western blotting results. However, compared with the OGD-NC mimic group or OGD-NC inhibitor group, the release was reduced or enhanced in the OGD-mimic group or OGD-inhibitor group (Fig 5A and 5B). After neutrophil isolation and purification, neutrophil transwell co-culture experiments confirmed that OGD-treated cell supernatants were able to increase neutrophil migration compared to normal culture. The OGD-mimic group showed a much lower number of cell migrations than the OGD-NC mimic group, whereas the OGD-inhibitor group showed the opposite trend (Fig 5C). Therefore, our study revealed that upregulation of miR-328-3p could prevent neutrophil migration by lowering IL-1β and IL-18 secretion in OGD-treated HUVECs, a process that may affect the NF-κB signaling pathway.

### 3.4 Overexpression of miR-328-3p improves neutrophil extracellular traps formation after OGD treatment

H3Cit, MPO, and NE are proteins that are released by neutrophils and are important in the composition of neutrophil extracellular traps [18]. We extracted cell supernatants from each

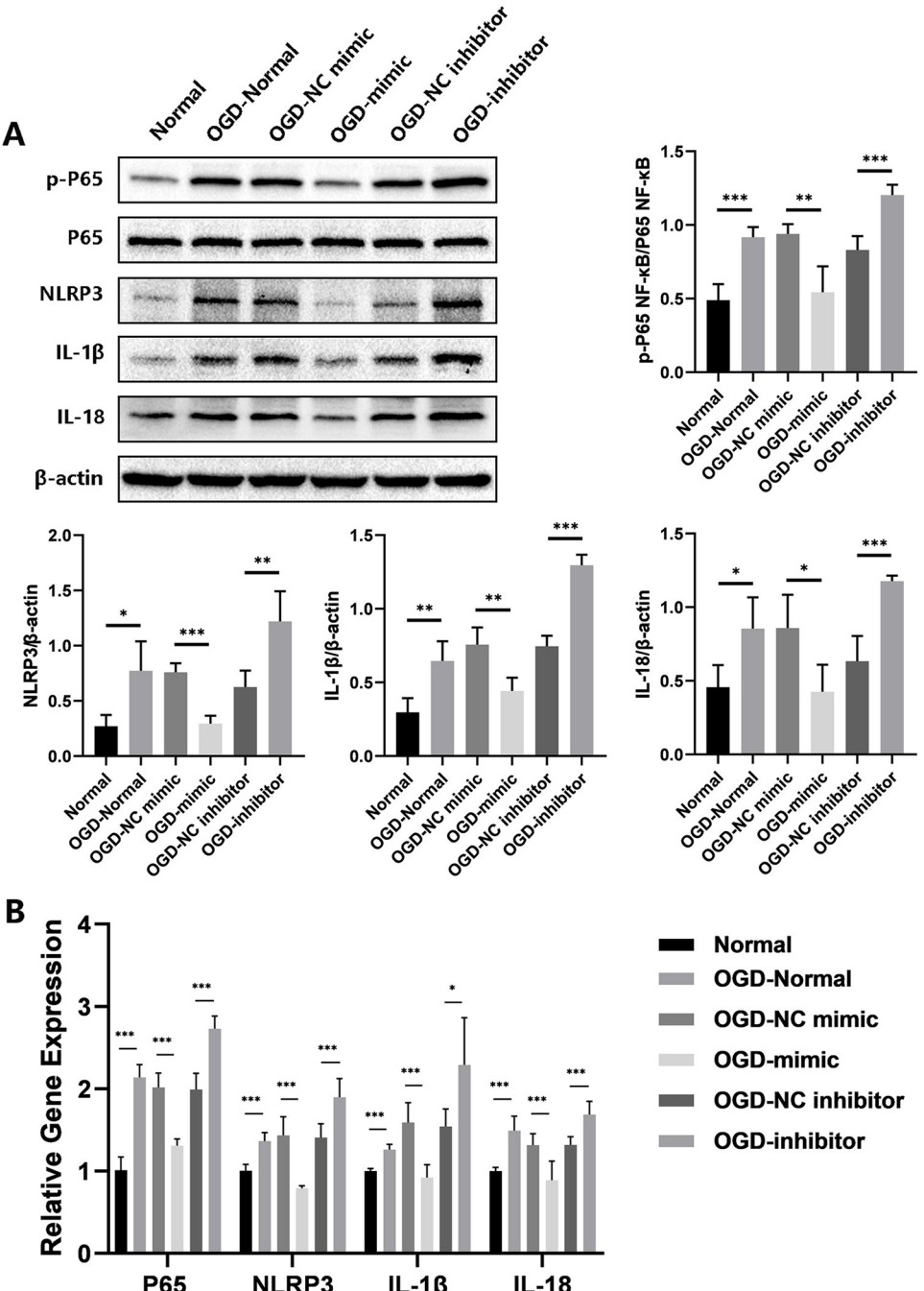

**Fig 4. miR-328-3p affects the expression of NF-κB signaling pathway and inflammation-related molecules after OGD treatment.** A. The protein expression levels of p-P65 NF-κB, NLRP3, IL-1β, and IL-18 were analyzed by Western blotting in each group (n = 4). B. The mRNA expression levels of P65 NF-κB, NLRP3, IL-1β, and IL-18 in each group were analyzed by qRT-qPCR (n = 3). Data are presented as the mean ± SD. *P < 0.05; **P < 0.01; ***P < 0.001.

group after OGD, co-cultured them with neutrophils, and localized NETs by immunofluorescence staining for H3Cit and MPO. We used Image J to calculate the number of cells co-localized with H3Cit and MPO as a percentage of the number of DAPI-stained cells. The results showed that the OGD-Normal group had more co-localization of H3Cit and MPO than the

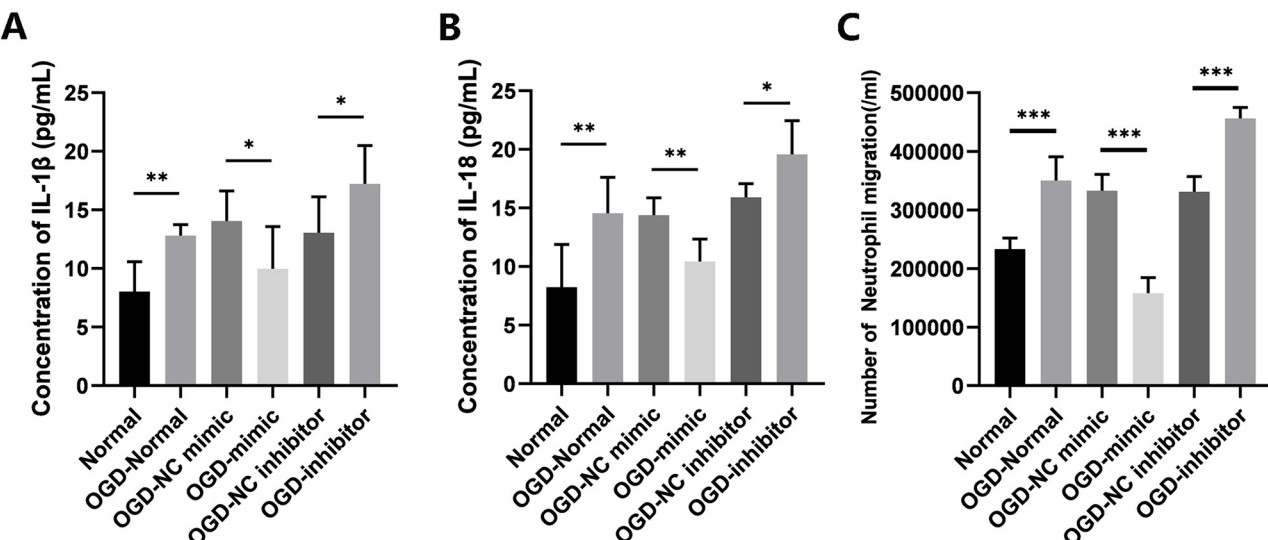

**Fig 5. Overexpression of miR-328-3p inhibits IL-1β and IL-18 release and neutrophil migration after OGD treatment.** A. Supernatant IL-1β levels were measured via ELISA. B. Supernatant IL-18 levels were measured via ELISA. C. Neutrophil migration counts. Data are presented as the mean ± SD (n = 3). *P < 0.05; **P < 0.01; ***P < 0.001.

Normal group. We draw the conclusion that, in contrast to the Normal group, more NETs were created in the OGD-Normal group. Meanwhile, the proportion of NETs was reduced in the OGD-mimic group compared to the OGD-NC mimic group, while the OGD-inhibitor group showed the opposite trend (Fig 6A and 6B). To further analyze the levels of NETs, we used ELISA to detect the levels of MPO and NE in the supernatants of co-cultured cells. According to the experimental findings, the OGD-Normal group had greater MPO and NE levels than the Normal group. However, compared with the OGD-NC mimic group or the OGD-NC inhibitor group, the release was reduced or enhanced in the OGD-mimic group or the OGD-inhibitor group (Fig 6C and 6D). We have found that miR-328-3p regulates the NETs formation by affecting inflammatory factors released by OGD-treated HUVECs.

## 4. Discussion

Patients with thrombotic diseases have severe burdens due to high rates of morbidity and mortality [19]. The surface of blood vessels is covered by endothelium, which separates blood from the vessel wall and sustains endovascular homeostasis [20]. When endothelial dysfunction or injury occurs, a variety of adverse consequences occur, such as increased inflammation, decreased vasodilation, and increased thrombosis, all of which contribute to the progression of thrombotic diseases [21]. It has been found that miRNAs have the potential to be used as therapeutic targets for cardiovascular diseases, as well as for use as novel biomarkers [22]. Notably, miR-328-3p can affect cancer progression [10, 11], and miR-328-3p also has a unique role in inflammation regulation. miR-328-3p is closely related to the progression of osteoarthritis [23], human pulmonary artery endothelial cell function [24], and airway inflammation [25]. Meanwhile, miR-328-3p was able to improve endothelial injury and inflammation. For example, miR-328-3p protected vascular endothelial cells from oxidized LDL-induced injury [26]. The above results led us to consider whether miR-328-3p has a protective effect against endothelial injury and is a potential target for endothelial injury. Therefore, we found that miR-328-3p was lowly expressed in HUVECs treated with OGD for the first time. The harmful

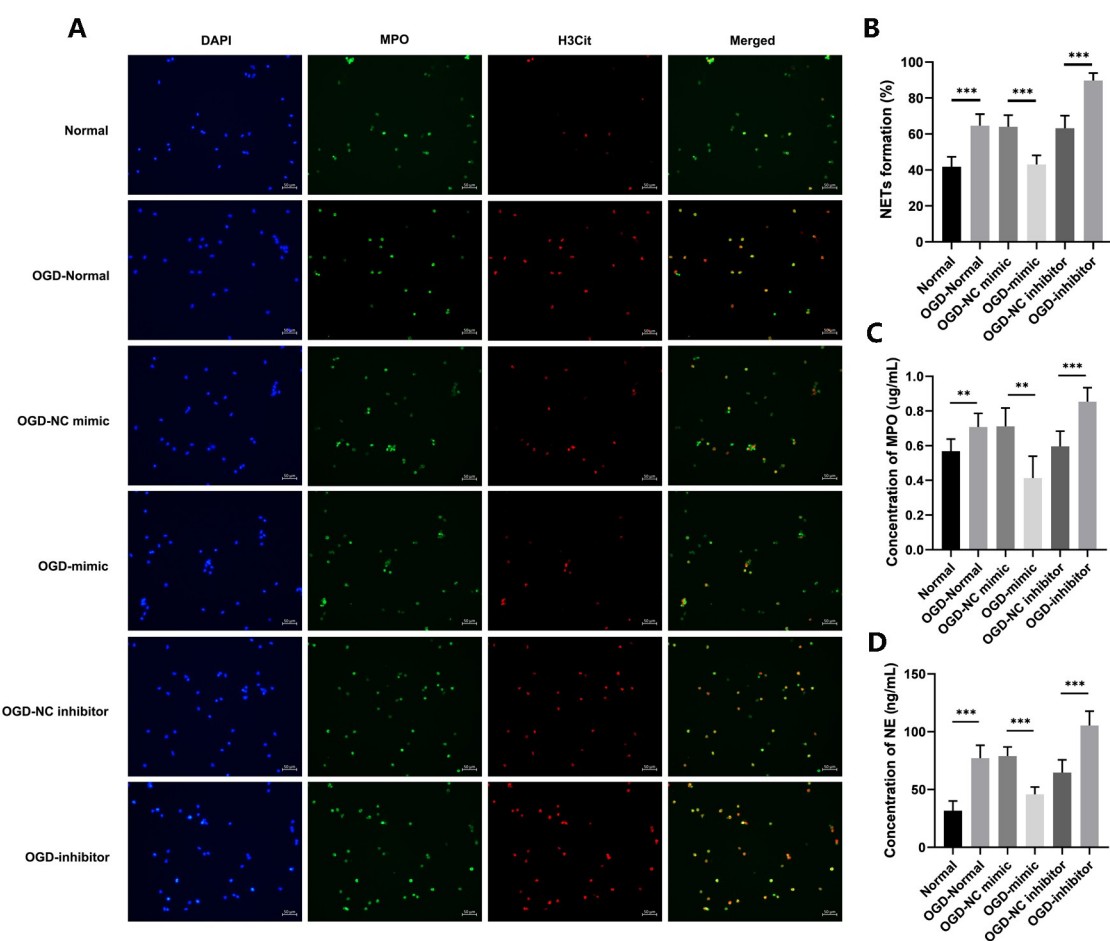

**Fig 6. Overexpression of miR-328-3p inhibits NETs formation after OGD treatment.** A. Localization of NETs by immunofluorescence staining for H3Cit and MPO. B. Percentage of the number of cells co-localized by H3Cit and MPO in the number of DAPI-stained cells. C and D. ELISA assay for determination of MPO and NE levels in supernatants. Data are presented as the mean ± SD (n = 3). **P < 0.01; ***P < 0.001.

effects of OGD were reversed and cell growth was encouraged by increased miR-328-3p expression. According to our findings, miR-328-3p might be crucial in defending endothelium cells while thrombotic illnesses are developing.

Upon endothelial activation, endothelial cells and neutrophils promote neutrophil recruitment through ligand-receptor interactions, and these interactions are critical for thrombus formation [27]. Cytokines released by endothelial cells (e.g., ROS, IL-8, and IL-1β) can accelerate NETs formation [28]. Meanwhile, enzymes with proteolytic activity released by NETs, such as elastase, metalloproteinase, protein hydrolase 3, and histone G, can increase the blood vessel's permeability by destroying the connections between endothelial cells [29]. Together, endothelial cell damage and NETs formation influence the progression of thrombotic disease. Consistent with these studies, our study found that OGD-treated cell supernatants promoted the formation of NETs in the middle by immunofluorescence localization of H3Cit and MPO, as evidenced by ELISA experiments detecting MPO and NE. However, miR-328-3p overexpression showed the reverse result. Therefore, we suggest that the NF-κB signaling pathway may be affected, which would result in decreasing inflammatory factor IL-1β and IL-18 release and reducing neutrophil migration and NETs formation.

Using a dual-luciferase reporter assay, we were able to determine that miR-328-3p directly binds to TLR2 in order to study its impact on endothelial injury and inflammation in OGD-treated endothelium. The expression of TLR2 was increased in OGD-induced HUVECs. In the OGD-mimic group, we found decreased TLR2 expression. TLR2 is one of the TLRs, and we also examined the expression of other TLRs such as TLR3, TLR4, and TLR9 (which are more studied in HUVECs [30–32]). The experimental results revealed that OGD treatment could affect the expression of TLR3, TLR4, and TLR9 in HUVECs, which again corroborated that TLRs are associated with inflammatory responses. However, after OGD treatment, there was no correlation trend of TLR3, TLR4, and TLR9 expression in HUVECs overexpressing and interfering with miR-328-3p. The above experiments prove the rationality of studying TLR2 under our experimental conditions. Meanwhile, NF-κB, as a downstream molecule of TLR2, is an inflammatory signaling pathway that acts as a crucial player in immune responses and cellular inflammatory responses. Inflammatory stimuli (e.g., IL-1β, ROS, LPS, TNFα, and thrombin) can activate NF-κB. Innate immunity and inflammation are caused by cellular genes being transcribed by NF-κB dimers, which bind DNA in the nucleus [33]. During acute inflammation, NF-κB promotes neutrophil apoptosis to counteract the body's inflammatory response [34, 35]. Through NF-κB activation, TLR2 can further activate Nod-like receptor protein-3 (NLRP3) [36, 37]. We learned that NLRP3 can regulate neutrophil migration and NETs activation by affecting caspase-1 causing the activation and cleavage of IL-18 and IL-1β [38, 39]. Our findings support earlier research showing that miR-328-3p targeted TLR2 to activate the NF-κB signaling pathway during OGD treatment. MiR-328-3p overexpression inhibited the NF-κB signaling pathway, decreased NLRP3 expression, IL-1β and IL-18 release, ameliorated cell injury, and the formation of NETs.

In addition to affecting endothelial cells, the critical role of NETs in thrombosis has been demonstrated. It has been discovered that NETs serve as the structural basis for thrombus development [39–41] and are crucial in immune thrombosis [42]. NETs are abundant in venous, arterial, and cancer-related thrombi [43–45]. In recent clinical studies, it has been shown that patients with DVT have significantly altered serum or plasma levels of NETs biomarkers, such as H3Cit and cell-free DNA (cfDNA), and that these might be predictive of DVT in conjunction with D-dimer [46, 47]. Thus, NETs not only provide different ideas for the pathogenesis of thrombotic diseases but also provide new potential references and directions for the prediction and diagnosis of thrombotic diseases. Our study used the oxygen-glucose deprivation method, which is a simulation of in vitro thrombogenic environment and ischemic hypoxic injury. MiR-328-3p had a protective role against HUVECs treated with OGD, and via the NF-κB signaling pathway, it influenced the release of inflammatory factors and NETs. In summary, our study is valuable for understanding the mechanism of endothelial injury in thrombosis, which is favorable for the development of biomarkers and novel drugs.

However, the current study still has some limitations. Both TLR2 and TLR4 are involved in noninfectious injury, and future studies of the role of TLR4 in OGD-treated HUVECs may provide valuable insights. We need to show that miR-328-3p functions in vivo and clarify the precise mechanism by which NETs promote thrombosis, as we have only looked into the role and mechanism of miR-328-3p in cellular tests.

## 5. Conclusions

To summarize, our study revealed that miR-328-3p expression was downregulated in OGD-treated HUVECs and miR-328-3p overexpression reversed the proliferative capacity of OGD-treated HUVECs. Mechanistically, miR-328-3p affects the NF-κB signaling pathway by

directly targeting TLR2. After OGD treatment, miR-328-3p overexpression may have anti-inflammatory effects by reducing inflammatory factors and NETs formation.

## Supporting information

**S1 Raw images.**
(PDF)

## Author Contributions

**Conceptualization:** Jianwen Mo.

**Data curation:** Mengting Yao.

**Formal analysis:** Jianwen Mo.

**Funding acquisition:** Tianting Guo, Jianwen Mo.

**Investigation:** Mengting Yao, Chucun Fang, Zilong Wang, Jian Wu.

**Methodology:** Mengting Yao, Chucun Fang, Zilong Wang.

**Project administration:** Tianting Guo, Jianwen Mo.

**Resources:** Chucun Fang, Zilong Wang, Dongwen Wu, Jiacheng Ma, Jian Wu.

**Software:** Chucun Fang, Zilong Wang.

**Supervision:** Tianting Guo, Dongwen Wu, Jiacheng Ma, Jianwen Mo.

**Validation:** Mengting Yao.

**Visualization:** Mengting Yao.

**Writing – original draft:** Mengting Yao.

**Writing – review & editing:** Mengting Yao, Jianwen Mo.

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
