## [Decision Letter · Decision Letter 0]

26 Dec 2023

PONE-D-23-38890miR-328-3p targets TLR2 to ameliorate oxygen-glucose deprivation injury and neutrophil extracellular trap formation in HUVECs via inhibition of the NF-κB signaling pathwayPLOS ONE

Dear Dr. Mo,

Thank you for submitting your manuscript to PLOS ONE. After careful consideration, we feel that it has merit but does not fully meet PLOS ONE’s publication criteria as it currently stands. Therefore, we invite you to submit a revised version of the manuscript that addresses the points raised during the review process.

Your manuscript was reviewed by two experts and both of them provided minor comments as provided below.

We look forward to receiving your revised manuscript.

Kind regards,

Partha Mukhopadhyay, Ph.D.

Section Editor

PLOS ONE

Journal Requirements:

3. Thank you for stating the following financial disclosure: "This work was supported by the National Natural Science Foundation of China (No. 82160375); Natural Science Foundation of Jiangxi Province (No. 20202BABL206035); Science and Technology Planning Project of Jiangxi Provincial Health Commission in 2023 (NO. 202312146); and Science and Technology Program of Jiangxi Provincial Administration of Traditional Chinese Medicine (Project No. 2021A374)."

Reviewers' comments:

Reviewer's Responses to Questions

**Comments to the Author**

1. Is the manuscript technically sound, and do the data support the conclusions?

Reviewer #1: Yes

Reviewer #2: Yes

2. Has the statistical analysis been performed appropriately and rigorously? 

Reviewer #1: No

Reviewer #2: I Don't Know

3. Have the authors made all data underlying the findings in their manuscript fully available?

Reviewer #1: Yes

Reviewer #2: Yes

4. Is the manuscript presented in an intelligible fashion and written in standard English?

Reviewer #1: Yes

Reviewer #2: Yes

5. Review Comments to the Author

Reviewer #1: The authors utilize an oxygen-glucose deprivation (OGD)-induced HUVEC model to investigate the role of miR-328-3p in endothelial cell injury pathogenesis. Their findings suggest that miR-328-3p targets TLR2 and suppresses inflammatory response to OGD treatment. However, further clarification is required in several aspects:

Major Concerns:

1. The rationale for choosing miR-328-3p as a potential target in endothelial injury needs elaboration.

2. While the authors mention TLR2 abundance in endothelial cells, additional experimental evidence (e.g., QPCR of different TLRs in HUVECs) is necessary to justify focusing solely on TLR2.

3. Given the involvement of both TLR2 and TLR4 in non-infectious injuries, and the identification of TLR4 as a potential miR-328-3p target by TargetScan, investigating TLR4 expression in OGD-treated HUVECs could provide valuable insights.

Minor Points:

1. The IL1b primer sequence requires verification.

2. Include scale bars for Figure 1E and Figure 5A for clarity.

3. Ensure precise sample size reporting in all experiments, particularly Figures 2 and 3. Discrepancies between figure legends and data points (e.g., Figure 4A-B and 5C-D) need clarification.

4. Provide catalogue numbers for reagents and kits used.

5. Clarify the OGD treatment protocol. The discrepancy between 30 minutes in the anaerobic incubator (Page 14, line 126) and 6 hours of OGD is confusing.

6. Provide details on the lentivirus and luciferase assay vectors used.

7. Clearly explain the differences between the WT and MUT 3’-UTR sequences in the dual-luciferase reporter assay.

8. Refine the title of Figure 1 to be more descriptive and informative.

9. Ensure consistent formatting for all figure titles.

Reviewer #2: The Manuscript is well written. I have a few minor comments for the authors:

1) The authors should include scale bars in the images in Fig 1E and Fig 5A.

2) In the method section, for some of the sections like cell counting Kit-8 assay, the authors have mentioned the vendors and how included the catalog number. It will be useful if the authors include the catalog number of the product as well. Please add the catalog number for several of the reagents in the method section.

6. PLOS authors have the option to publish the peer review history of their article (what does this mean?). If published, this will include your full peer review and any attached files.

Reviewer #1: No

Reviewer #2: No

---

## [Author Response · Author response to Decision Letter 0]

20 Jan 2024

Dear Editors and Reviewers:

Thank you for your decision and construction comments on my manuscript. These comments are valuable and helpful for improving our manuscript. According to the editor and reviewers’ comments, we made some changes to the manuscript. In this revised version, changes to our manuscript were all highlighted within the " Revised Manuscript with Track Changes" file by using red-colored text. The responses to the reviewers’ comments are as follows:

Reviewer #1: The authors utilize an oxygen-glucose deprivation (OGD)-induced HUVEC model to investigate the role of miR-328-3p in endothelial cell injury pathogenesis. Their findings suggest that miR-328-3p targets TLR2 and suppresses inflammatory response to OGD treatment. However, further clarification is required in several aspects:

Major Concerns:

1. The rationale for choosing miR-328-3p as a potential target in endothelial injury needs elaboration.

Response: We sincerely appreciate the valuable comment. miR-328-3p has been much studied in various cancer progression. However, by reviewing various studies we found that miR-328-3p has a unique role in inflammation regulation. miR-328-3p is closely related to the progression of osteoarthritis[1], human pulmonary artery endothelial cell function[2], and airway inflammation[3]. Meanwhile, miR-328-3p can improve endothelial damage and inflammation. miR-328-3p protects vascular endothelial cells from oxidized LDL-induced injury[4]. It has been shown that miR-328 ameliorates LDL-induced injury and inflammation in HUVECs[5]. The above results made us consider whether miR-328-3p has a protective effect against endothelial injury and is a potential target for endothelial injury. Meanwhile, through experimental studies, we found that miR-328-3p expression was reduced in OGD-treated HUVECs. Overexpression of miR-328-3p improved OGD-induced proliferation, inflammatory factor expression, and NF-κB phosphorylation in HUVECs. In summary, miR-328-3p is a potential target for endothelial injury through theoretical and experimental studies. We have made changes and additions to the relevant parts of the article (Line 387-395).

2. While the authors mention TLR2 abundance in endothelial cells, additional experimental evidence (e.g., QPCR of different TLRs in HUVECs) is necessary to justify focusing solely on TLR2.

Response: We sincerely appreciate the valuable comment. A review of various studies revealed that TLR3[6, 7], TLR4[8, 9] and TLR9[10, 11] are expressed in HUVECs. Little or no TLR5 expression was observed in HUVECs[12], and TLR7 and TLR8 were not expressed in HUVECs[7]. There is no relevant study of TLR6 in HUVECs. Therefore, under the influence of OGD and miR-328-3p, we supplemented qRT-PCR experiments to detect the expression of TLR3, TLR4, and TLR9. The experimental results are shown below. OGD treatment could affect the expression of TLR3, TLR4, and TLR9 in HUVECs, again corroborating the effect of the inflammatory response on TLRs. However, after OGD treatment, there was no correlation trend of TLR3, TLR4, and TLR9 expression in HUVECs overexpressing and interfering with miR-328-3p. In conclusion, our experiments again demonstrated that miR-328-3p targets TLR2 in OGD-treated HUVECs and that miR-328-3p is not associated with other TLRs. Therefore, it is reasonable to focus on TLR2 only in our experimental study. Supplementary experiments and content we have added to the text (Line 295-299, 418-425, Figure 3).

3. Given the involvement of both TLR2 and TLR4 in non-infectious injuries, and the identification of TLR4 as a potential miR-328-3p target by TargetScan, investigating TLR4 expression in OGD-treated HUVECs could provide valuable insights.

Response: We sincerely appreciate the valuable comment. By reviewing the literature, we found that miR-328-3p targeting high mobility group box 1 (HMGB1) was able to affect the TLR4/NF-κB pathway[3]. However, we did not find relevant studies of miR-328-3p targeting TLR4 in HUVECs. Although TargetScan identified TLR4 as a potential miR-328-3p target, we could still verify it by dual luciferase reporter assay. Meanwhile, our qRT-PCR experiments revealed that TLR4 expression was upregulated in OGD-treated HUVECs. The above content is a potential direction for our study and something we need to address in our subsequent experiments. We have added this content as a limitation of our study in the discussion (Line 454-456). Once again, we thank the reviewers for their valuable comments

References

[1] Yang J, Zhang M, Yang D, et al. m(6)A-mediated upregulation of AC008 promotes osteoarthritis progression through the miR-328-3p‒AQP1/ANKH axis. Exp Mol Med. 2021. 53(11): 1723-1734.

[2] Hong L, Ma X, Liu J, et al. Circular RNA-HIPK3 regulates human pulmonary artery endothelial cells function and vessel growth by regulating microRNA-328-3p/STAT3 axis. Pulm Circ. 2021. 11(2): 20458940211000234.

[3] Cao X, Wang K, Zhu H. Yanghepingchuan granule improves airway inflammation by inhibiting autophagy via miRNA328-3p/high mobility group box 1/Toll-like receptor 4 targeting of the pathway of signaling in rat models of asthma. J Thorac Dis. 2023. 15(11): 6251-6264.

[4] Qin X, Guo J. MicroRNA-328-3p Protects Vascular Endothelial Cells Against Oxidized Low-Density Lipoprotein Induced Injury via Targeting Forkhead Box Protein O4 (FOXO4) in Atherosclerosis. Med Sci Monit. 2020. 26: e921877.

[5] Wu CY, Zhou ZF, Wang B, Ke ZP, Ge ZC, Zhang XJ. MicroRNA-328 ameliorates oxidized low-density lipoprotein-induced endothelial cells injury through targeting HMGB1 in atherosclerosis. J Cell Biochem. 2019. 120(2): 1643-1650.

[6] Guo Z, Chen L, Zhu Y, et al. Double-stranded RNA-induced TLR3 activation inhibits angiogenesis and triggers apoptosis of human hepatocellular carcinoma cells. Oncol Rep. 2012. 27(2): 396-402.

[7] Tissari J, Sirén J, Meri S, Julkunen I, Matikainen S. IFN-alpha enhances TLR3-mediated antiviral cytokine expression in human endothelial and epithelial cells by up-regulating TLR3 expression. J Immunol. 2005. 174(7): 4289-94.

[8] Liu X, Lu B, Fu J, Zhu X, Song E, Song Y. Amorphous silica nanoparticles induce inflammation via activation of NLRP3 inflammasome and HMGB1/TLR4/MYD88/NF-kb signaling pathway in HUVEC cells. J Hazard Mater. 2021. 404(Pt B): 124050.

[9] Huang L, Li Y, Cheng Z, Lv Z, Luo S, Xia Y. PCSK9 Promotes Endothelial Dysfunction During Sepsis Via the TLR4/MyD88/NF-κB and NLRP3 Pathways. Inflammation. 2023. 46(1): 115-128.

[10] Li B, Zhang R, Li J, et al. Antimalarial artesunate protects sepsis model mice against heat-killed Escherichia coli challenge by decreasing TLR4, TLR9 mRNA expressions and transcription factor NF-kappa B activation. Int Immunopharmacol. 2008. 8(3): 379-89.

[11] Li Y, Ou K, Wang Y, Luo L, Chen Z, Wu J. TLR9 agonist suppresses choroidal neovascularization by restricting endothelial cell motility via ERK/c-Jun pathway. Microvasc Res. 2022. 141: 104338.

[12] Hijiya N, Miyake K, Akashi S, Matsuura K, Higuchi Y, Yamamoto S. Possible involvement of toll-like receptor 4 in endothelial cell activation of larger vessels in response to lipopolysaccharide. Pathobiology. 2002. 70(1): 18-25.

Minor Points:

1. The IL1b primer sequence requires verification.

Response: We thank you for the helpful comments and suggestions. After our check, the primer sequence for IL-1β was written incorrectly. The correct primer sequence of IL-1β is in the table below. At the same time, we made changes in Table 3.

 Sequence (5’ to 3’)

Forward primer ATGATGGCTTATTACAGTGGCAA

Reverse primer GTCGGAGATTCGTAGCTGGA

2. Include scale bars for Figure 1E and Figure 5A for clarity.

Response: We sincerely appreciate the valuable comment. We have added scale bars to the images in Figure 1E and Figure 6A.

3. Ensure precise sample size reporting in all experiments, particularly Figures 2 and 3. Discrepancies between figure legends and data points (e.g., Figure 4A-B and 5C-D) need clarification.

Response: 

We sincerely appreciate the valuable comment. In Fig. 2, the sample size is 3 for Dual-luciferase reporter assay and qRT-qPCR and 4 for Western blotting. In Fig. 3, the sample size was 3 for qRT-qPCR and 4 for Western blotting. The sample size for all experiments in Fig. 1, Fig. 4 and Fig. 5 was 3. We have marked in the article. 

The differences in the release of IL-1β and IL-18 between the groups are mainly demonstrated in Figure 4A-B. The release of IL-1β and IL-18 was more in the OGD-Normal group than in the Normal group, the release of IL-1β and IL-18 was less in the OGD-mimic group than in the OGD-NC mimic group, and the release of IL-1β and IL-18 was more in the OGD-inhibitor group than in the OGD-NC inhibitor group. The main reason for these differences is that miR-328-3p targeting TLR2 affects the NF-κB signaling pathway, leading to a decrease in the release of IL-1β and IL-18. The differences in cellular supernatant MPO and NE between the groups are mainly shown in Figure 5C-D while reacting to the differences in NETs formation. The results showed that more NETs were formed in the OGD-Normal group than the Normal group, less NETs were formed in the OGD-mimic group than the OGD-NC mimic group, and more NETs were formed in the OGD-inhibitor group than the OGD-NC inhibitor group. This reflects the effect of miR-328-3p on NETs formation after OGD treatment. The sample size of all Elisa experiments was 3, and the duplicate well was 1. Figures 4A-B and 5C-D were incorrectly selected as the chart type, so we changed the chart type to make the Figure clearer (Figures 5A-B and 6C-D). Thanks again to the reviewers for their valuable comments.

4. Provide catalogue numbers for reagents and kits used.

Response: We thank you for the helpful comments and suggestions. In Chapter '2. Materials and methods', the relevant reagents and kits are labeled with catalogue numbers (Line 105-247).

5. Clarify the OGD treatment protocol. The discrepancy between 30 minutes in the anaerobic incubator (Page 14, line 126) and 6 hours of OGD is confusing.

Response: We sincerely appreciate the valuable comment. After the anaerobic incubator reached 94% N2, 5% CO2, and 1% O2 conditions before the start of the OGD experiment, we placed each group of cells into the anaerobic incubator. During the process of opening the anaerobic incubator, oxygen enters the incubator and prevents the incubator from maintaining the 1% O2 concentration. After several OGD experiments, we found that the oxygen concentration was about 7-8% after the cells were put into the incubator, and it took 30 minutes for the oxygen concentration to drop to 1%. Therefore, the 6h OGD was officially started when the groups of cells were in 1% oxygen concentration. We have added relevant content to Chapter '2.3. Oxygen-glucose deprivation (OGD) model' (Line 128-131).

6. Provide details on the lentivirus and luciferase assay vectors used.

Response: We thank you for the helpful comments and suggestions. Lentivirus and luciferase assay vectors are listed below. We have added relevant content to Chapter '2.2. Cell transfection' (Line 120-123) and Chapter '2.7. Dual-luciferase reporter assay' (Line 166-171).

Lentivirus vectors and sequences:

 Vector name Sequence (5’ to 3’)

miR-328-3p mimics LV3(H1/GFP&Puro)- hsa-miR-328-3p CTGGCCCTCTCTGCCCTTCCGT

negative control mimics LV3(H1/GFP&Puro)-NC TTCTCCGAACGTGTCACGT

miR-328-3p inhibitor LV3(H1/GFP&Puro)-hsa-miR-328-3p inhibitor ACGGAAGGGCAGAGAGGGCCAG

negative control inhibitor LV3(H1/GFP&Puro)-NC TTCTCCGAACGTGTCACGT

Luciferase assay vectors and sequences:

 Vector name Sequence (5’ to 3’)

hsa-TLR2-miR-328-3p-WT psiCHECKTM-2-Vector ggttgacttcatggatgcagaacccatggatatagAGGGCCAactgtaatctgtagcaactggcttagt

hsa-TLR2-miR-328-3p-MUT psiCHECKTM-2-Vector ggttgacttcatggatgcagaacccatggatatagTCCCGGTactgtaatctgtagcaactggcttagt

7. Clearly explain the differences between the WT and MUT 3’-UTR sequences in the dual-luciferase reporter assay.

Response: We sincerely appreciate the valuable comment. The difference between the WT and MUT 3'-UTR sequences is that the WT is base complementary paired with the miRNA and the MUT is not complementary paired with the miRNA. We have added the relevant content in Chapter '2.7. Dual-luciferase reporter assay' (Line 166-167).

8. Refine the title of Figure 1 to be more descriptive and informative.

Response: We sincerely appreciate the valuable comment. We have changed the title to "Fig. 1. miR-328-3p expression is down-regulated and promotes cell proliferation in OGD-treated HUVECs" (Line 277-278).

9. Ensure consistent formatting for all figure titles.

Response: We thank you for the helpful comments and suggestions. All figure titles have been bolded. 

Reviewer #2: The Manuscript is well written. I have a few minor comments for the authors:

1) The authors should include scale bars in the images in Fig 1E and Fig 5A.

Response: We sincerely appreciate the valuable comment. We have added scale bars to the images in Figure 1E and Figure 6A.

2) In the method section, for some of the sections like cell counting Kit-8 assay, the authors have mentioned the vendors and how included the catalog number. It will be useful if the authors include the catalog number of the product as well. Please add the catalog number for several of the reagents in the method section.

Response: We thank you for the helpful comments and suggestions. In Chapter '2. Materials and methods', the relevant reagents and kits are labeled with catalogue numbers (Line 105-247).

We tried our best to improve the manuscript and made some changes in the manuscript. We appreciate for Editor's and Reviewers' warm work earnestly and hope the correction will meet with approval.

Thank you very much for your attention and consideration!

Yours sincerely,

Mengting Yao

---

## [Decision Letter · Decision Letter 1]

9 Feb 2024

miR-328-3p targets TLR2 to ameliorate oxygen-glucose deprivation injury and neutrophil extracellular trap formation in HUVECs via inhibition of the NF-κB signaling pathway

PONE-D-23-38890R1

Dear Dr. Mo,

We’re pleased to inform you that your manuscript has been judged scientifically suitable for publication and will be formally accepted for publication once it meets all outstanding technical requirements.

Kind regards,

Partha Mukhopadhyay, Ph.D.

Section Editor

PLOS ONE

Additional Editor Comments (optional):

Reviewers' comments:

Reviewer's Responses to Questions

**Comments to the Author**

1. If the authors have adequately addressed your comments raised in a previous round of review and you feel that this manuscript is now acceptable for publication, you may indicate that here to bypass the “Comments to the Author” section, enter your conflict of interest statement in the “Confidential to Editor” section, and submit your "Accept" recommendation.

Reviewer #1: All comments have been addressed

2. Is the manuscript technically sound, and do the data support the conclusions?

Reviewer #1: Yes

3. Has the statistical analysis been performed appropriately and rigorously? 

Reviewer #1: Yes

4. Have the authors made all data underlying the findings in their manuscript fully available?

Reviewer #1: Yes

5. Is the manuscript presented in an intelligible fashion and written in standard English?

Reviewer #1: Yes

6. Review Comments to the Author

Reviewer #1: (No Response)

7. PLOS authors have the option to publish the peer review history of their article (what does this mean?). If published, this will include your full peer review and any attached files.

Reviewer #1: No

---

## [Editor Report · Acceptance letter]

15 Feb 2024

PONE-D-23-38890R1 

PLOS ONE

Dear Dr. Mo, 

I'm pleased to inform you that your manuscript has been deemed suitable for publication in PLOS ONE. Congratulations! Your manuscript is now being handed over to our production team.

Kind regards, 

on behalf of

Dr. Partha Mukhopadhyay 

Section Editor

PLOS ONE